# Effect of SARS-CoV-2 proteins on vascular permeability

Rossana Rauti[1†], Meishar Shahoha[2,3†], Yael Leichtmann-Bardoogo[1†], Rami Nasser[4], Eyal Paz[2,3], Rina Tamir[1], Victoria Miller[1], Tal Babich[1,2], Kfir Shaked[1,2], Avner Ehrlich[5], Konstantinos Ioannidis[5], Yaakov Nahmias[5], Roded Sharan[4], Uri Ashery[2,3,6], Ben Meir Maoz[1,3,6]*

[1]Department of Biomedical Engineering, Tel Aviv University, Tel Aviv, Israel; [2]School of Neurobiology, Biochemistry and Biophysics, The George S. Wise Faculty of Life Sciences, Tel Aviv University, Tel Aviv, Israel; [3]Sagol School of Neuroscience, Tel Aviv University, Tel Aviv, Israel; [4]Blavatnik School of Computer Science, Tel Aviv University, Tel Aviv, Israel; [5]Grass Center for Bioengineering, The Hebrew University of Jerusalem, Jerusalem, Israel; [6]The Center for Nanoscience and Nanotechnology, Tel Aviv University, Tel Aviv, Israel

*For correspondence:
Bmaoz@tauex.tau.ac.il

†These authors contributed equally to this work

Competing interest: The authors declare that no competing interests exist.

**Abstract** Severe acute respiratory syndrome (SARS)-CoV-2 infection leads to severe disease associated with cytokine storm, vascular dysfunction, coagulation, and progressive lung damage. It affects several vital organs, seemingly through a pathological effect on endothelial cells. The SARS-CoV-2 genome encodes 29 proteins, whose contribution to the disease manifestations, and especially endothelial complications, is unknown. We cloned and expressed 26 of these proteins in human cells and characterized the endothelial response to overexpression of each, individually. Whereas most proteins induced significant changes in endothelial permeability, nsp2, nsp5_c145a (catalytic dead mutant of nsp5), and nsp7 also reduced CD31, and increased von Willebrand factor expression and IL-6, suggesting endothelial dysfunction. Using propagation-based analysis of a protein–protein interaction (PPI) network, we predicted the endothelial proteins affected by the viral proteins that potentially mediate these effects. We further applied our PPI model to identify the role of each SARS-CoV-2 protein in other tissues affected by coronavirus disease (COVID-19). While validating the PPI network model, we found that the tight junction (TJ) proteins cadherin-5, ZO-1, and β-catenin are affected by nsp2, nsp5_c145a, and nsp7 consistent with the model prediction. Overall, this work identifies the SARS-CoV-2 proteins that might be most detrimental in terms of endothelial dysfunction, thereby shedding light on vascular aspects of COVID-19.

## Introduction

Coronavirus disease (COVID-19) caused by the 2019 novel coronavirus (2019-nCoV/SARS-CoV-2) led to a global pandemic in 2020. By late September 2021, coronavirus had infected more than 220 million people worldwide, causing over 4.5 million deaths. After the initial phase of the viral infection, ~ 30 % of patients hospitalized with COVID-19 develop severe disease with progressive lung damage, known as severe acute respiratory syndrome (SARS), and a severe immune response. Interestingly, additional pathologies have been observed, such as hypoxemia and cytokine storm which, in some cases, lead to heart and kidney failure, and neurological symptoms. Recent observations suggest that these pathologies are mainly due to increased coagulation and vascular dysfunction (*Lee et al., 2021*; *Libby and Lüscher, 2020*; *Siddiqi et al., 2020*). It is currently believed that in addition to being a respiratory disease, COVID-19 might also be a 'vascular disease' (*Lee et al., 2021*), as it may result in a leaky vascular barrier and increased expression of von Willebrand factor (VWF) (*Siddiqi et al.,*

*2020*), responsible for increased coagulation, cytokine release, and inflammation (*Siddiqi et al., 2020*; *Teuwen et al., 2020*; *Aid et al., 2020*; *Potus et al., 2020*; *Wazny et al., 2020*; *Pum et al., 2021*; *Barbosa et al., 2021*; *Lin et al., 2020*; *Matarese et al., 2020*; *Xiao et al., 2020*). Recent studies suggest that the main mechanism disrupting the endothelial barrier occurs in several stages: First, a direct effect on the endothelial cells that causes an immune response of the vascular endothelium (endotheliitis) and endothelial dysfunction. Second, lysis and death of the endothelial cells *Teuwen et al., 2020*; *Xiao et al., 2020* followed by sequestering of human angiotensin I-converting enzyme 2 (hACE2) by viral spike proteins that activate the kallikrein–bradykinin and renin–angiotensin pathways, increasing vascular permeability (*Teuwen et al., 2020*; *Varga et al., 2020*). Last, overreaction of the immune system, during which a combination of neutrophils and immune cells producing reactive oxygen species, inflammatory cytokines (e.g., interleukin [IL]-1β, IL-6, and tumor necrosis factor), and vasoactive molecules (e.g., thrombin, histamine, thromboxane A2, and vascular endothelial growth factor), and the deposition of hyaluronic acid lead to disruption of endothelial junctions, increased vascular permeability, and leakage and coagulation (*Libby and Lüscher, 2020*; *Teuwen et al., 2020*; *Varga et al., 2020*). Of great interest is the effect on the brain's vascular system. Cerebrovascular effects have been suggested to be among the long-lasting effects of COVID-19. Indeed, the susceptibility of brain endothelial cells to direct SARS-CoV-2 infection was found to increase due to increased expression of hACE2 in a blood flow-dependent manner, leading to a unique gene expression process that might contribute to the cerebrovascular effects of the virus (*Pober and Sessa, 2007*).

While many studies point out the importance of the vascular system in COVID-19 (*Kaneko et al., 2021*; *Jung et al., 2020b*; *Nägele et al., 2020*), only a few *Pons et al., 2020*; *Chioh et al., 2020*; *Nascimento Conde et al., 2020*; *Buzhdygan et al., 2020* have looked at the direct vascular response to the virus. Most of those reports stem from either clinical observations, or in vitro studies or in vivo

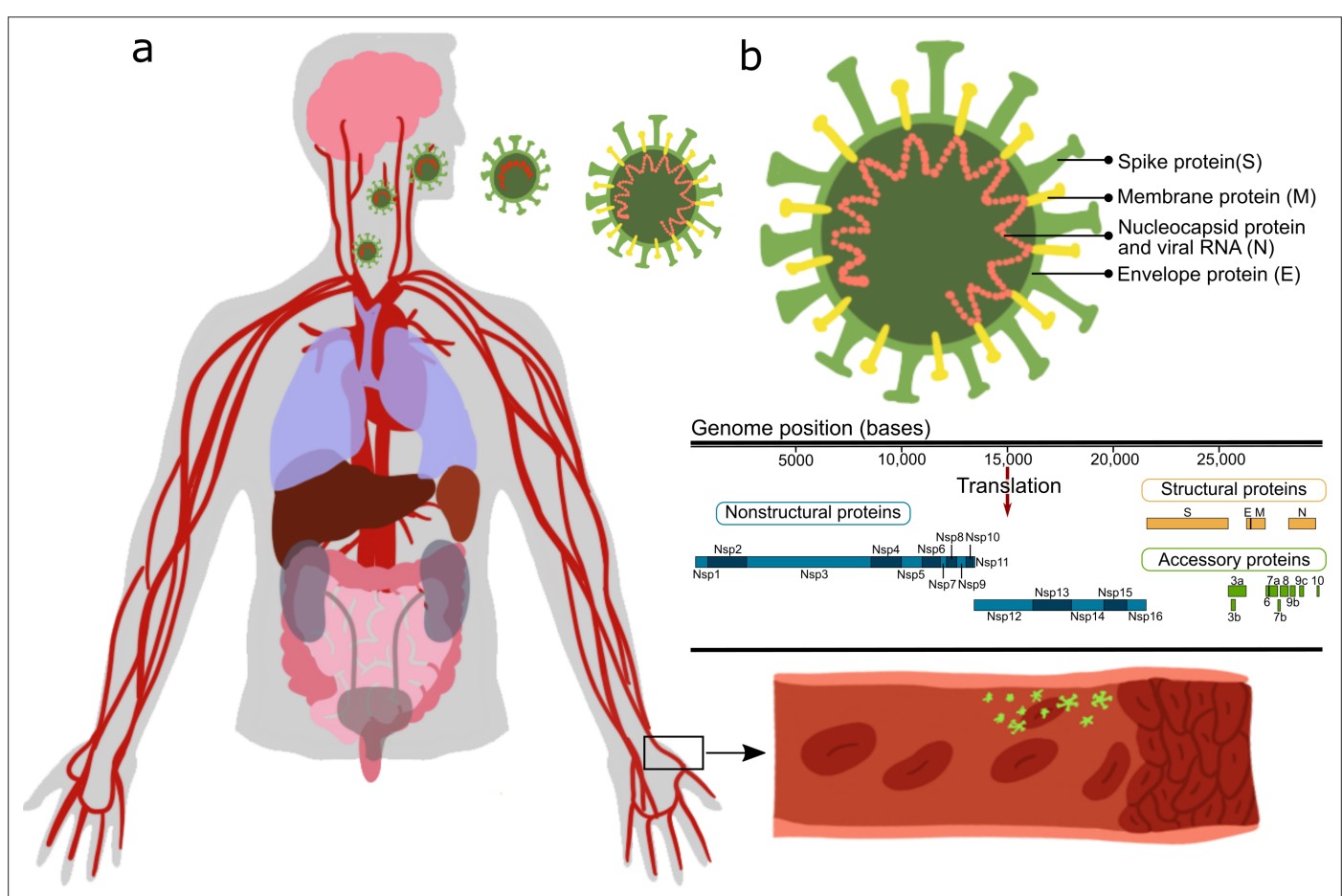

**Figure 1.** Effect of severe acute respiratory syndrome (SARS)-CoV-2 proteins on endothelial cells. (a) Sketch representing the main organs affected by SARS-CoV-2; (**b**) structure and gene composition of SARS-CoV-2.

studies in which animals/cells were transfected with the SARS-CoV-2 virus and their systemic cellular response assessed, without pinpointing the specific viral protein(s) causing the observed changes. SARS-CoV-2 is an enveloped virus with a positive-sense, single-stranded RNA genome of ~30 kb, encoding 29 proteins (*Figure 1*). These proteins can be classified as: *structural proteins*: S (spike proteins), E (envelope proteins), M (membrane proteins), N (nucleocapsid protein and viral RNA); *nonstructural proteins*: nsp1–16; *open reading frame accessory proteins*: orf3–10 (*Kim et al., 2020*; *Hu et al., 2021*). *Table 1* summarizes the known effects of specific SARS-CoV-2 proteins (*Gordon et al., 2020*; *Peng et al., 2020b*; *Procko, 2020*; *Cornillez-Ty et al., 2009*; *Romano et al., 2020*; *Hillen et al., 2020*; *Chi et al., 2003*). The functionality of some of these is still unknown. Moreover, a considerable knowledge gap still exists regarding molecular mechanisms, especially the protein–protein interaction (PPI) pathways (*Cowen et al., 2017*), leading to tissue dysfunction.

To tackle these challenges, we cultured human umbilical vein endothelial cells (HUVECs) and systematically transduced them with lentiviral particles encoding 26 out of the 29 viral proteins, separately. The three remaining genes were not included in this study purely for technical reasons. We then examined their effects on HUVEC monolayer permeability and the expression of factors involved in vascular permeability and coagulation. The results were analyzed in the context of virus–host and host–host PPI networks. By combining the insights from the experimental and computational results, we generated a model that explains how each of the 26 proteins of SARS-CoV-2, including a mutated form of nsp5, the catalytic dead mutant termed nsp5_c145a, affects the protein network regulating vascular functionality. Moreover, once the PPI model was validated with our experimental data, we applied it to more than 250 proteins that have been identified in the literature as affected by the SARS-CoV-2 proteins. This enabled us to pinpoint the more dominant SARS-CoV-2 proteins and chart their effects. Overall, this work shows how each of the SARS-CoV-2 proteins differentially affects vascular functionality; moreover, once the model was validated, we applied it to identify how SARS-CoV-2 proteins interact with proteins that have been significantly correlated with changes in cell functionality.

## Results
### SARS-CoV-2 proteins impair barrier properties affecting cell-junction proteins

Increasing numbers of studies indicate a significant role for the vasculature in the physiological response to SARS-CoV-2. However, neither the exact molecular mechanism that leads to these effects nor the individual contribution of any of the SARS-CoV-2 proteins is known. Plasmids encoding SARS-CoV-2 proteins were cloned into lentivirus vectors, with eGFP-encoding vector used as a negative control. To shed light on the vascular response to the virus, HUVECs were cultured on different platforms, transduced with these lentiviral particles, and assessed for the effects of the virus proteins on different functionalities. Culturing HUVEC on Transwells or glass coverslips (*Figure 2a*) allowed us to identify how the specific proteins affect endothelial functionality. To ensure proper infection, the control vector included a GFP label, which enabled us to estimate infection efficiency at around 70 % (*Figure 2a*). Since the most basic function of the endothelium is to serve as a barrier, we sought to identify the changes in endothelium permeability in response to the SARS-CoV-2 proteins, and to pinpoint which of these proteins have the most significant effect. Barrier functions and properties were measured via trans-epithelial-endothelial electrical resistance (TEER), a standard method that identifies changes in impedance values, reflecting the integrity and permeability of the cell monolayer (*Srinivasan et al., 2015*). The GFP control and nine SARS-CoV-2 proteins did not show any significant change in TEER values (compared to the untreated condition), whereas 18 of the SARS-CoV-2 proteins caused significant changes in value (see plot in *Figure 2b*). The most dominant permeability changes were observed with nsp5_c145a, nsp13, nsp7, orf7a, and nsp2, with a 20–28% decrease in TEER values (*Figure 2—figure supplement 1*, and *Figure 2c*), in which the different SARS-CoV-2 proteins are listed and the gradual color change from red to violet represents the progressive reduction in TEER values. *Figure 2—figure supplement 1* shows the comparison in TEER values before the infection and 3 and 4 days after the infection, showing that the permeability changes in the cells exposed to the viral proteins are maintained.

Next, we analyzed some of the proteins that exhibited the most significant (nsp2, nsp5_c145a, and nsp7) or least significant (S) changes in TEER value for changes in expression of the cell-junction

**Table 1.** Severe acute respiratory syndrome (SARS)-CoV-2 proteins.

| SARS-CoV-2 proteins | General impact |
| --- | --- |
| **Structural proteins** | |
| S (spike) | Spike protein, mediates binding to ACE2, fusion with host membrane<br>Surface glycoprotein, needs to be processed by cellular protease TMPRSS2 (*Gordon et al., 2020*) |
| M (membrane) | Membrane glycoprotein, the predominant component of the envelope<br>A major driver for virus assembly and budding (*Gordon et al., 2020*) |
| E (envelope) | Envelope protein, involved in virus morphogenesis and assembly<br>Coexpression of M and E is sufficient for virus-like particle formation and release (*Gordon et al., 2020*) |
| N (nucleocapsid) | Nucleocapsid phosphoprotein binds to RNA genome (*Gordon et al., 2020*) |
| **Nonstructural proteins** | |
| nsp1 | Leader sequence, suppresses host antiviral response<br>Antagonizes interferon induction to suppress host antiviral response (*Gordon et al., 2020*) |
| nsp2 | Interferes with host cell signaling, including cell cycle, cell-death pathways, and cell differentiation<br>May serve as an adaptor for nsp3<br>Not essential for virus replication, but deletion of nsp2 diminishes viral growth and RNA synthesis (*Gordon et al., 2020*; *Procko, 2020*) |
| nsp3 | nsp3–nsp4–nsp6 complex involved in viral replication<br>Functions as papain-like protease (*Gordon et al., 2020*) |
| nsp4 | nsp3–nsp4–nsp6 complex involved in viral replication (*Gordon et al., 2020*)<br>The complex is predicted to nucleate and anchor viral replication complexes on double-membrane vesicles in the cytoplasm (mitochondria) |
| nsp5 | Inhibits interferon I signaling processes by intervening in the NF-$\kappa$B process and breaking down STAT one transcription factor<br>Functions as 3-chymotrypsin-like protease, cleaves the viral polyprotein (*Gordon et al., 2020*) |
| nsp5_c145a | Catalytic dead mutant of nsp5 (*Gordon et al., 2020*) |
| nsp6 | nsp3–nsp4–nsp6 complex involved in viral replication<br>Limits autophagosome expansion<br>Components of the mitochondrial complex V (the complex regenerates ATP from ADP) copurify with nsp6 (*Gordon et al., 2020*) |
| nsp7 | Cofactor of nsp12<br>nsp7–nsp8 complex in part of RNA polymerase (nsp7, 8, 12 – replication complex)<br>Affects electron transport, GPCR signaling, and membrane trafficking (*Gordon et al., 2020*; *Peng et al., 2020b*; *Romano et al., 2020*; *Hillen et al., 2020*) |
| nsp8 | Cofactor of nsp12<br>nsp7–nsp8 complex in part of RNA polymerase. Affects the signal recognition particle and mitochondrial ribosome (*Gordon et al., 2020*; *Peng et al., 2020b*; *Romano et al., 2020*; *Chi et al., 2003*) |
| nsp9 | ssRNA binding protein (can bind both DNA and RNA, but prefers ssRNA)<br>Interacts with the replication complex (nsp7, 8, 12) (*Cornillez-Ty et al., 2009*) |
| nsp10 | Cofactor of nsp16 and nsp14 (*Romano et al., 2020*)<br>Essential for nsp16 methyltransferase activity (stimulator of nsp16)<br>Zinc finger protein essential for replication (*Gordon et al., 2020*; *Peng et al., 2020b*) |
| nsp11 | Unknown function |
| nsp12 | Functions as an RNA-direct RNA polymerase, the catalytic subunit<br>Affects the spliceosome (*Gordon et al., 2020*; *Peng et al., 2020b*; *Romano et al., 2020*; *Hillen et al., 2020*) |
| nsp13 | Has helicase and 5' triphosphatase activity<br>Initiates the first step in viral mRNA capping nsp13,14,16 installs the cap structure onto viral mRNA in the cytoplasm instead of in the nucleus, where the host mRNA is capped (*Gordon et al., 2020*; *Peng et al., 2020b*; *Romano et al., 2020*; *Ivanov et al., 2004*) |

*Table 1 continued on next page*

*Table 1 continued*

| SARS-CoV-2 proteins | General impact |
| --- | --- |
| nsp14 | In addition to the capping function of the methyltransferase, nsp14 is also an endonuclease (3'–5' exoribonuclease) that corrects mutations during genome replication (*Gordon et al., 2020*; *Peng et al., 2020b*; *Romano et al., 2020*) |
| nsp15 | Endoribonuclease has uridine-specific endonuclease activity, essential for viral RNA synthesis (*Gordon et al., 2020*; *Romano et al., 2020*) |
| nsp16 | May involve complexation with nsp10 and nsp14, for stabilization of homoenzyme, for capping the mRNA (*Gordon et al., 2020*; *Peng et al., 2020b*; *Romano et al., 2020*) |
| Open reading frame (accessory factors) | |
| orf3a | Packaging into virions<br>Mediates trafficking of spike protein by providing ER/golgi retention signals<br>Induces IL-6b, activates NF-$\kappa$B, activates the NLRP3 inflammasome (*Gordon et al., 2020*) |
| orf3b | Interferon antagonist and involved in pathogenesis (*Gordon et al., 2020*) |
| orf6 | Type I interferon antagonist, suppresses the induction of interferon, and interferon signaling pathways (*Gordon et al., 2020*) |
| orf7a | May be related to viral-induced apoptosis (*Gordon et al., 2020*) |
| orf7b | Unknown function |
| orf8 | Recombination hotspot<br>Induces ER stress and activates NLRP3 inflammasomes<br>Low similarity to SAR-CoV (*Gordon et al., 2020*) |
| orf9b | Suppresses host antiviral response<br>Targets the mitochondrion-associated adaptor molecules MAVS and limits host cell interferon responses (*Gordon et al., 2020*) |
| orf9c | No evidence that this protein is expressed during SARS-CoV-2 infection (*Gordon et al., 2020*) |
| orf10 | No evidence that this protein is expressed during SARS-CoV-2 infection (*Gordon et al., 2020*) |

proteins such as CD31 (*Figure 2d and e*), cadherin 1–5, occludin, and ZO 1–3 (presented later), indicating altered barrier functions. Analysis of the immunocytochemistry (ICC) (*Figure 2d and e*) showed, as expected, that nsp2, nsp5_c145a, and nsp7 significantly reduce the expression levels of CD31 compared to the untreated, eGFP, and S conditions, suggesting a deterioration in barrier function. Hence, these data show a differential effect of SARS-CoV-2 proteins on endothelial functionality and provide a mechanistic explanation for the reduction in endothelial integrity.

## Increased endothelial inflammatory response caused by SARS-CoV-2 proteins

It is known that SARS-CoV-2 can cause a severe cytokine storm (*Pum et al., 2021*; *Wang et al., 2020a*) and a significant increase in coagulation-related pathologies. As we were interested in identifying the role of the vasculature in these observations, we stained and analyzed the expression level of VWF (*Figure 3a and b*), which is highly correlated with coagulation (*Rietveld et al., 2019*). Similar to the CD31 staining, we characterized only those proteins that resulted in a significant decrease in TEER values (nsp2, nsp5_c145a, and nsp7). As shown in *Figure 3a and b*, the control samples did not exhibit marked expression of VWF, whereas the cells transfected with nsp2, nsp5_c145a, and nsp7 showed a significant change in VWF expression. Moreover, as VWF is also associated with increased inflammation (*Kawecki et al., 2017*), we monitored changes in cytokine expression due to the different SARS-CoV-2 proteins (*Figure 3c*). We were particularly interested in IL-6, which has been identified as one of the most dominant cytokines expressed due to SARS-CoV-2 infection (*Wang et al., 2020a*; *Akbari and Rezaie, 2020*; *Peruzzi et al., 2020*; *Liao et al., 2020b*; *Liao et al., 2020a*). We observed that 13 out of the 26 proteins caused an increase in IL-6 secretion, 3 of which had resulted in a decrease in barrier function and increased VWF expression.

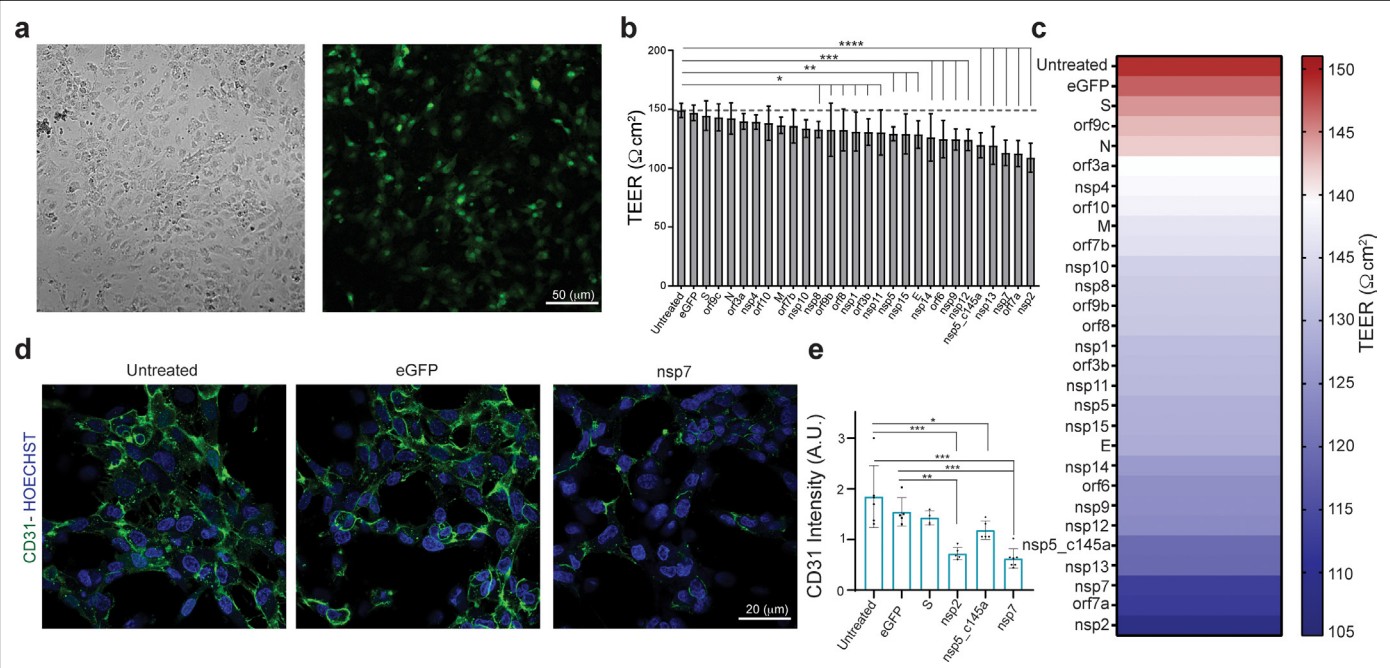

**Figure 2.** Effect of severe acute respiratory syndrome (SARS)-CoV-2 proteins on human umbilical vein endothelial cell (HUVEC) at day 3. (a) Bright-field and fluorescent image of infected eGFP HUVEC, scale bar: 50 μm; (b) changes in barrier functions as a result of SARS-CoV-2 proteins were assessed by trans-epithelial-endothelial electrical resistance (TEER) measurement. Note the statistical differences compared to the untreated control condition, assessed by F-statistic with two-way ANOVA test, followed by the Holm–Sidak test for multiple comparisons; (c) color map showing a gradual decrease in TEER values compared to the untreated condition at day 3; (d) immunocytochemistry (ICC) for CD31 (green) and Hoechst (blue) for the three specified conditions, scale bar: 20 μm; (e) analysis of CD31 expression levels.

The online version of this article includes the following figure supplement(s) for figure 2:

**Figure supplement 1.** Effect of severe acute respiratory syndrome (SARS)-CoV-2 proteins on human umbilical vein endothelial cell (HUVEC) functionality.

## Correlation between vascular permeability impairment and viral proteins

We then investigated how SARS-CoV-2 causes the observed changes in HUVECs permeability. We collected sets of proteins responsible for specific functionalities of endothelial cells. We also constructed an integrated viral–host and host–host PPI network. For each viral protein and each prior functional set, we measured the network proximity between the viral protein and the human functional set using a network propagation algorithm. We scored the significance of these propagation calculations by comparing them to those obtained on random PPI networks with the same node degrees. Proteins receiving high and significant scores were most likely to interact with the specific SARS-CoV-2 protein and thus might cause the observed functional changes. When comparing the overall effects of the 26 SARS-CoV-2 proteins on endothelial TJ proteins (e.g., cadherin 1–5, occludin, and ZO 1–3), we found a correlation between the effects of the SARS-CoV-2 proteins and TEER values (*Figure 4a*). Moreover, some of the proteins that significantly affected the TEER parameters (*Figure 2c*) were also observed to be significantly proximal to the permeability-related set. These included nsp2, nsp7, and nsp13 (*Figure 4a*). Our algorithm identified cadherin-2, α-catenin, β-catenin, δ-catenin, and ZO 1 and 2 as the most susceptible proteins to SARS-CoV-2 infection (*Figure 4b*).

To validate our PPI network model, we performed immunostaining of some TJ proteins (β-catenin, cadherin-5, ZO-1, and occludin) of HUVEC transfected with viral proteins and to compare it to the model prediction. Similar to the CD31 staining, we characterized only those proteins that significantly decreased TEER values (nsp2, nsp5_c145a, and nsp7) compared to the eGFP and untreated condition (*Figure 5*). As shown in *Figure 5a–c*, the cells transfected with nsp2, nsp5_c145a, and nsp7 showed a significant reduction in the β-catenin, cadherin-5, and ZO-1 intensity, confirming the ability of the SARS-CoV-2 proteins to impair endothelial permeability.

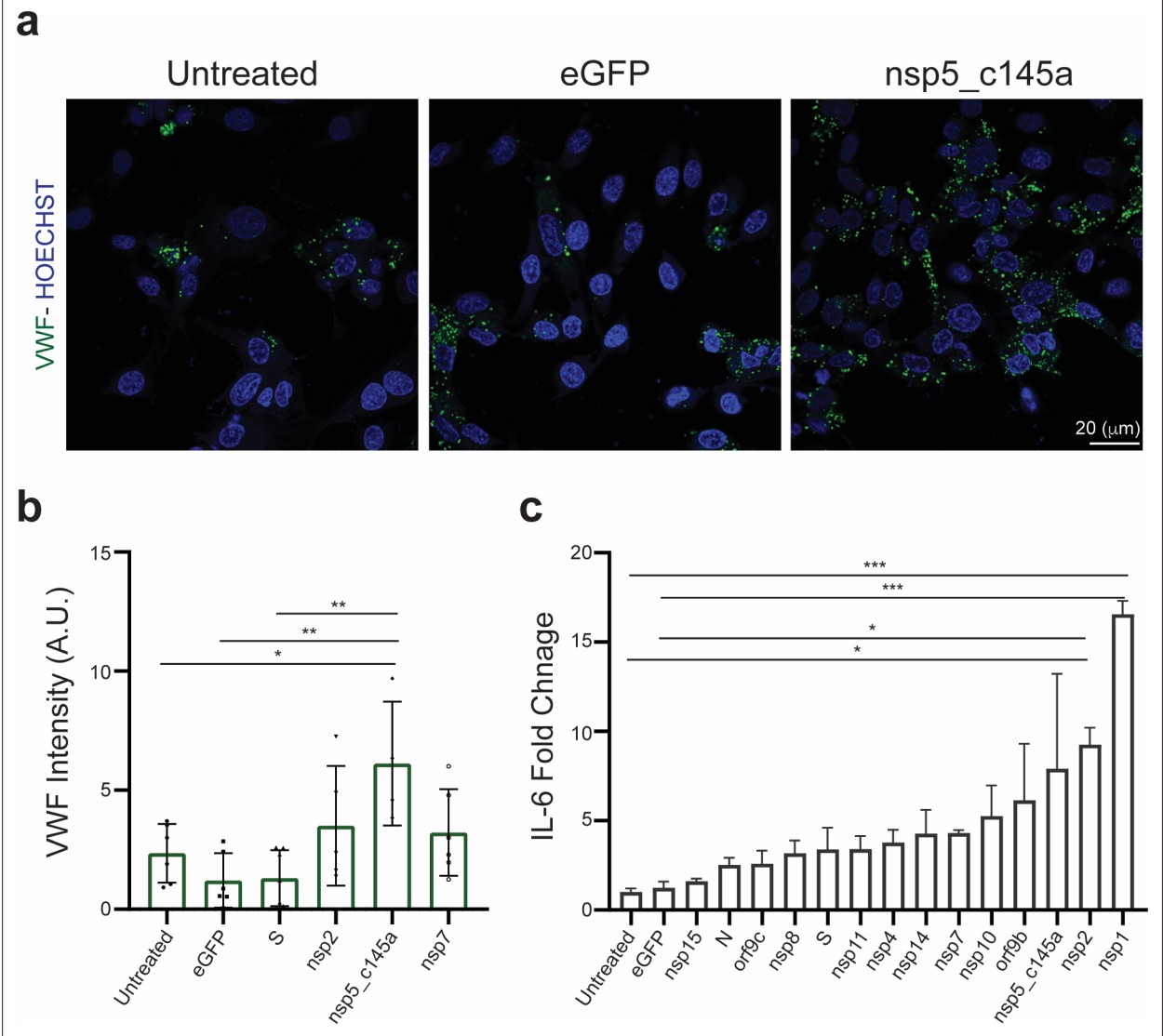

**Figure 3.** Human umbilical vein endothelial cell (HUVEC) response to specific proteins. (a) Confocal reconstructions of HUVEC stained for von Willebrand factor (VWF) (green) and Hoechst (blue) for three conditions: control (untreated), eGFP, and nsp5_c145a, scale bar: 20 µm; (b) analysis of VWF expression levels; (c) fold change of interleukin (IL)-6 in response to the different proteins.

Once the model was validated, we used it to identify how the individual SARS-Cov-2 proteins affect nine other different vascular endothelial cells. As a starting point, we created a table (*Table 2*) (based on the literature) where we compared the expression of 12 different TJ proteins across nine different types of endothelial cells. We then applied the network-based model to identify which endothelial cells are more susceptible to the different SARS-Cov-2 proteins. As can be seen in *Figure 6*, there are significant differences in the response of various viral proteins on different types of vascular endothelial cells. For example,, the viral proteins nsp13, nsp11, orf6, and S seem to have a significant effect on the different types of vascular endothelial cells, according to the network score detected. However, the proteins m, E, n, nsp12, and nsp8 are the less interactive with the vascular cells.

As our network propagation model is highly correlated with our experimental results, we applied it to other physiological systems that are known to be affected by SARS-CoV-2. We created a list of all proteins that are known to be affected by the SARS-CoV-2 proteins according to the literature (*Supplementary file 1A*, white columns). The table was composed of both proteins identified experimentally via western blot, proteomics, and immunohistochemistry (marked in blue) and those identified clinically as being highly correlated with loss of specific functionality in specific tissues (marked

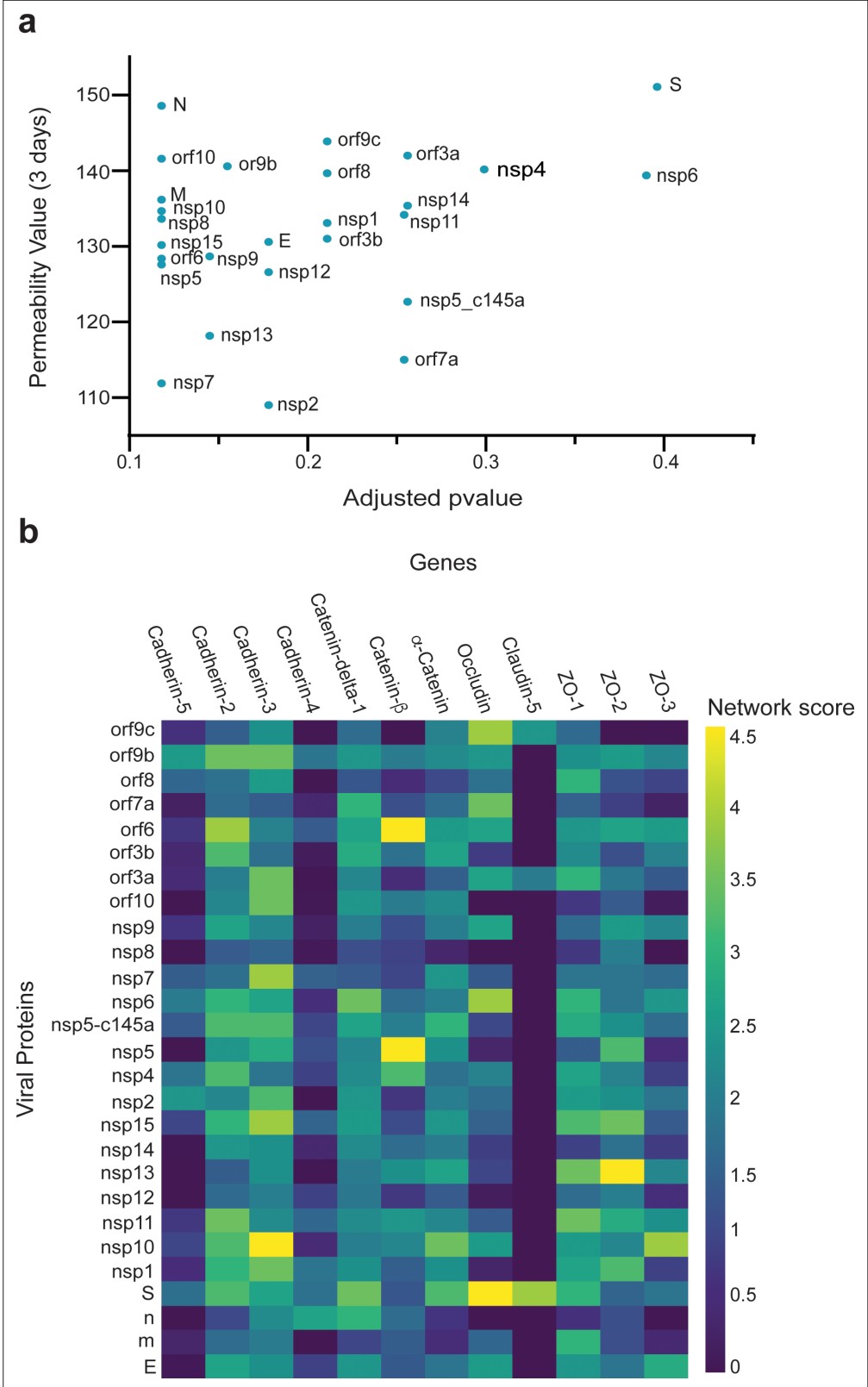

**Figure 4.** Correlation between viral protein effect on permeability and proximity to permeability-related proteins in a protein–protein interaction (PPI) network. (a) Correlation of adjusted p-value versus permeability (Pearson = 0.295); (b) proximity between vascular proteins and the viral proteins.

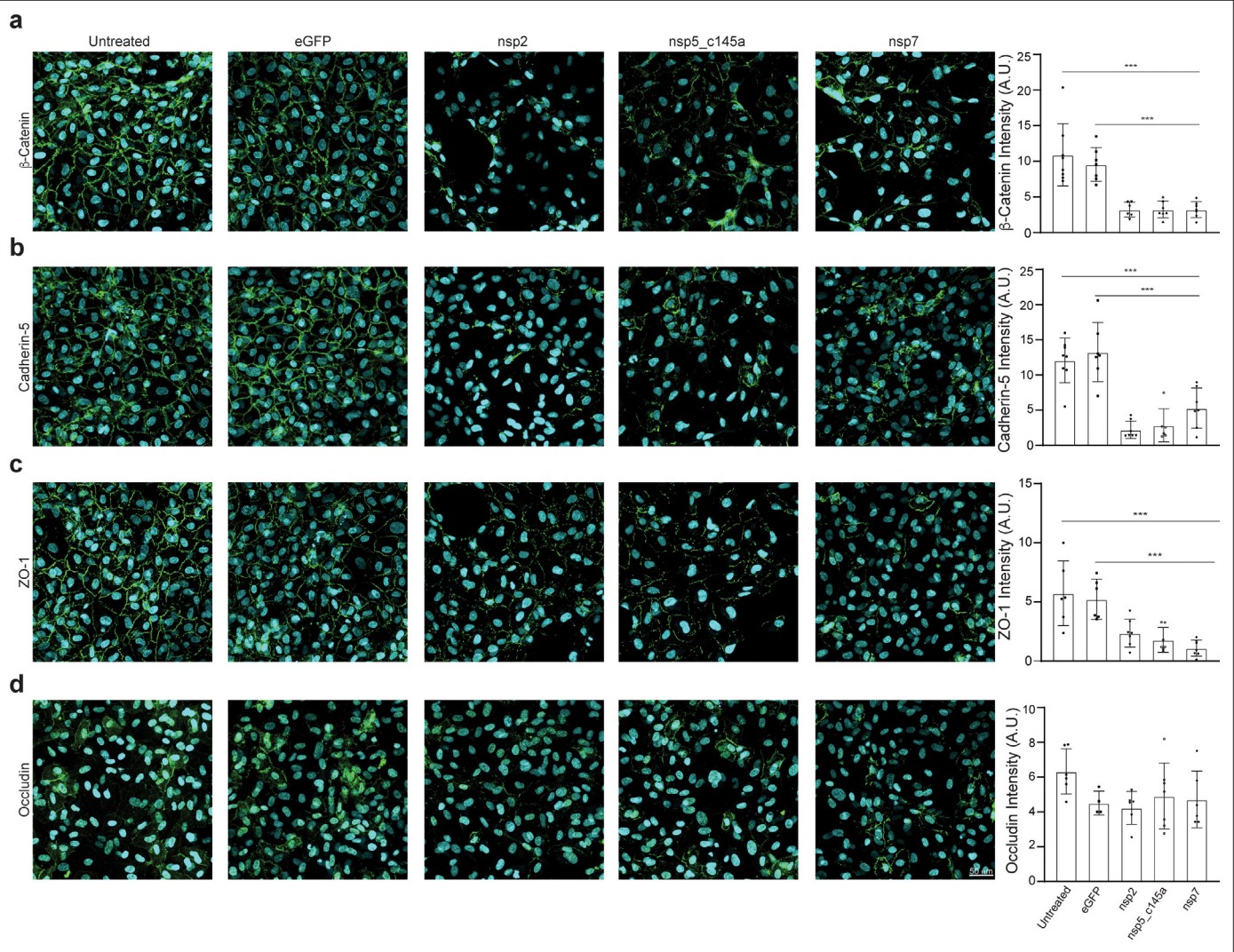

**Figure 5.** Tight-junctions impairment by severe acute respiratory syndrome (SARS)-CoV-2 proteins. Immunocytochemistry (ICC) for (a) $\beta$-catenin (green), (b) cadherin-5 (green), (c) ZO-1 (green), (d) occludin (green), and Hoechst (blue) for five specified conditions: untreated, eGFP, nsp2, nsp5-c1451, and nsp7. Relative quantification is shown for each tight junction (TJ) protein for all five conditions on the right. Scale bar: 50 μm.

in red). We then applied the network-based model to identify which proteins in *Supplementary file 1A* are most susceptible to the different SARS-CoV-2 proteins. As can be seen in *Figure 7—figure supplements 2–7*, *Supplementary file 1A and B*, specific SARS-CoV-2 proteins were identified as affecting specific proteins in specific tissues. As expected, most of the SARS-CoV-2 proteins affected more than one protein, the most salient being nsp11, nsp4, and nsp7 (*Figure 7b*), each of which was predicted to affect more than 40 different proteins. An additional parameter that should be considered is the protein's 'distance' from the viral proteins. This value represents the number of hops in the PPI network from a given protein to the viral proteins, where a value of 1 represents a direct viral–host connection. We hypothesized that the closer the distance between the viral proteins and the given protein, the more significant the viral effect. *Supplementary file 1A* (gray columns) and *Figure 7c* present the calculated distances. Most of the identified proteins in *Supplementary file 1A* were classified with a distance of 1 or 2 from the virus, suggesting more severe putative effects. A very clear example, are the T cells, macrophages, lung epithelial and cardiomyocytes which show that the most significant effect was by the viral proteins which present a close connection with the relative cell proteins reported. This suggest a potential effect on the related functional or metabolic pathway (*Supplementary file 1A*).

**Table 2.** Comparison of tight junction (TJ) proteins expression among different types of vascular endothelial cells.

| Endothelial cells type | TJ proteins | | | | | | | | | | | |
|---|---|---|---|---|---|---|---|---|---|---|---|---|
| | Cadherin-2 | Cadherin-3 | Cadherin-4 | Cadherin-5 | δ-1-Catenin | β-Catenin | α-Catenin | Occludin | Claudin-5 | ZO-1 | ZO-2 | ZO-3 |
| Human pulmonary artery endothelial cells (HPAECs) (*Nakato et al., 2019*; *Chi et al., 2003*; *Ivanov et al., 2004*; *Ferreri et al., 2008*; *DiStefano et al., 2014*; *Zebda et al., 2013*; *Yuan et al., 2012*; *Wang et al., 2011*) | + | − | − | + | + | + | + | + | + | + | + | − |
| Human umbilical vein endothelial cells (HUVECs) (*Nakato et al., 2019*; *Chi et al., 2003*; *Ferreri et al., 2008*; *Wu et al., 2008*; *Dean et al., 2009*; *Polus et al., 2006*; *DeBusk et al., 2010*; *Wessells et al., 2009*) | + | + | + | + | + | + | + | + | + | + | + | + |
| Human umbilical artery endothelial cells (HUAECs) (*Nakato et al., 2019*; *Chi et al., 2003*; *Davis et al., 2003*; *Ikuno et al., 2017*; *Kevil et al., 1998*; *Kluger et al., 2013*) | − | + | + | + | + | + | − | + | + | + | + | + |
| Human great saphenous vein endothelial cells (HGSVECs) (*Nakato et al., 2019*; *Chi et al., 2003*; *Latif et al., 2006*; *Murakami et al., 2008*) | − | − | − | + | + | − | + | + | + | + | + | + |
| Human common carotid artery endothelial cells (HCCaECs) (*Nakato et al., 2019*; *Chi et al., 2003*) | + | − | + | − | − | + | − | + | + | + | + | + |
| Human aortic endothelial cells (HAoECs) (*Nakato et al., 2019*; *Chi et al., 2003*; *Wu et al., 2017*; *Sandig et al., 1999*; *DeMaio et al., 2006*) | − | − | − | − | + | + | + | + | + | + | + | + |
| Human coronary artery endothelial cells (HCAECs) (*Nakato et al., 2019*; *Chi et al., 2003*; *Wessells et al., 2009*; *Wu et al., 2004*; *Pinto et al., 2018*) | − | − | + | + | + | + | + | + | + | + | + | + |
| Human endocardial cells (HENDCs) (*Nakato et al., 2019*; *Chi et al., 2003*; *Vestweber et al., 2009*; *Bao et al., 2017*) | + | − | + | − | + | + | + | + | + | + | − | − |
| Human renal artery endothelial cells (HRAECs) (*Nakato et al., 2019*; *Chi et al., 2003*; *Maciel et al., 2018*) | − | − | + | + | − | + | − | + | + | + | − | − |

## Discussion

Due to the impact of SARS-CoV-2, many studies have looked at the physiological responses to the virus (*Lee et al., 2021*; *Libby and Lüscher, 2020*; *Siddiqi et al., 2020*; *Teuwen et al., 2020*; *Chioh et al., 2020*). In this work, we sought to identify how specific SARS-CoV-2 proteins affect the vasculature by assessing the effect of individual SARS-CoV-2 proteins on endothelial cells (HUVEC). This

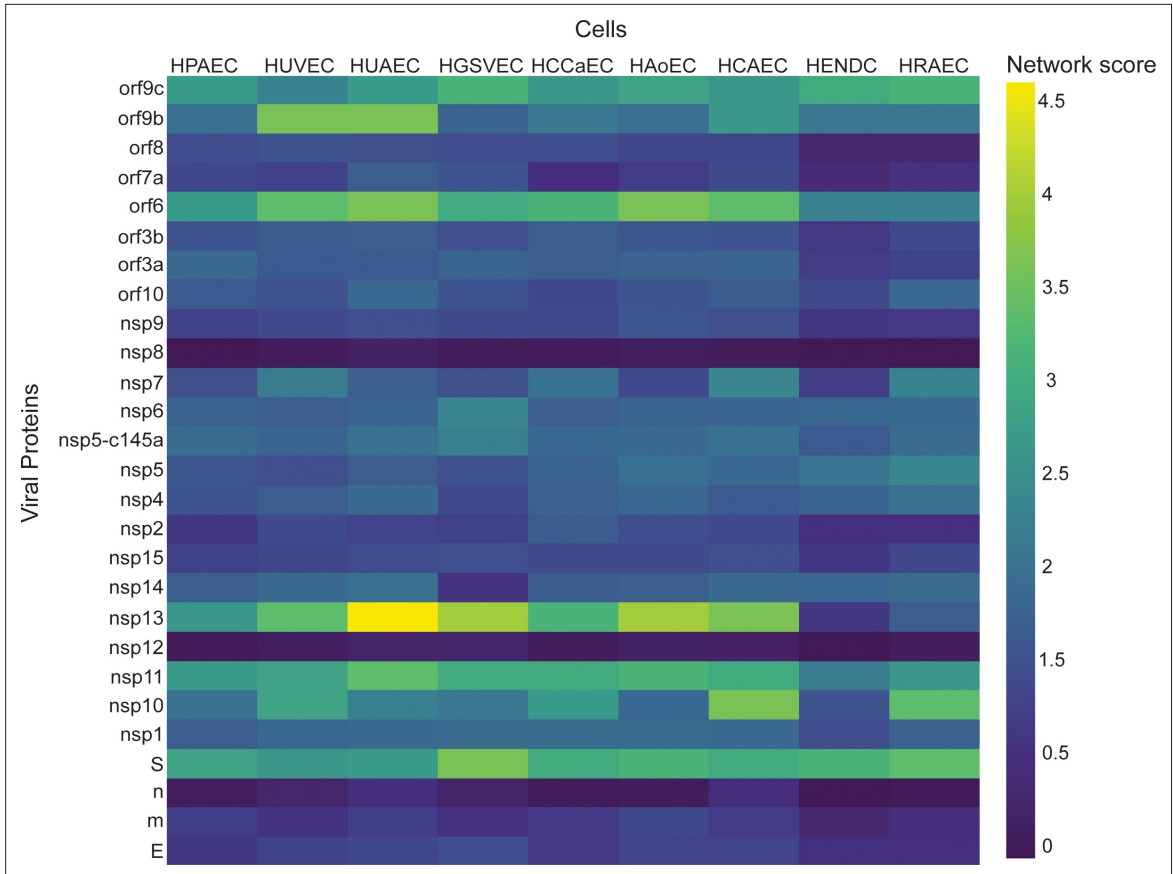

**Figure 6.** Correlation between viral proteins and different types of vascular endothelial cells. Correlation of adjusted p-value between vascular proteins identified in vascular endothelial cells and the viral proteins.

approach has significant advantages: it enables pinpointing and isolating how each of the SARS-CoV-2 proteins independently affects the endothelial response, and directly measuring endothelial functionality. The HUVEC model, derived from the umbilical cord, is physiologically representative of the human vascular endothelium, allowing the study of the physiological and pathological conditions as well as the effects of novel drugs on human endothelium (*Bouïs et al., 2001*; *Medina-Leyte et al., 2020*). Among technical advantages, cultured HUVECs are a simple in vitro vascular endothelial model, particularly suitable for studying endothelial properties and dynamics as well as the putative role of adhesion molecules, the synthesis of extracellular proteins and blood vessel maturation (*Vailhé et al., 2001*).

The current study showed that almost 70 % (18 out of 26) of the SARS-CoV-2 proteins affect endothelial barrier integrity; however, the most significant proteins were nsp2, nsp5_c145a, and nsp7, which also induced upregulated expression of the coagulation factor VWF and cytokine release. These critical facts can shed light on the multiple pathologies observed in SARS-CoV-2 infection, including cytokine storm, increased coagulation and related diseases (e.g., heart attack and stroke) (*Lee et al., 2021*; *Aid et al., 2020*), increased cardiovascular disease, and increased neurological symptoms. The results presented here showed an effect of in vitro cultured endothelial cells, which may lead to vasculature leakiness, consequently causing exotoxicity (i.e., the penetration of toxic reagents from the blood into the brain). While there are many parameters associated with functional changes, the use of advanced tools, including network-based analysis, enabled us to elucidate the specific proteins and the specific interactions that are predicted to cause these changes. The PPI network enabled us to predict that the changes observed in barrier function are possibly due to interactions between host proteins such as cadherin 2, α-catenin, β-catenin, δ-catenin, and ZO 1 and 2, and at least with the viral proteins nsp2, nsp5_c145a, and nsp7. Moreover, we validated our PPI model performing further immunostaining analysis demonstrating not only the ability of the viral proteins to strongly impair TJ

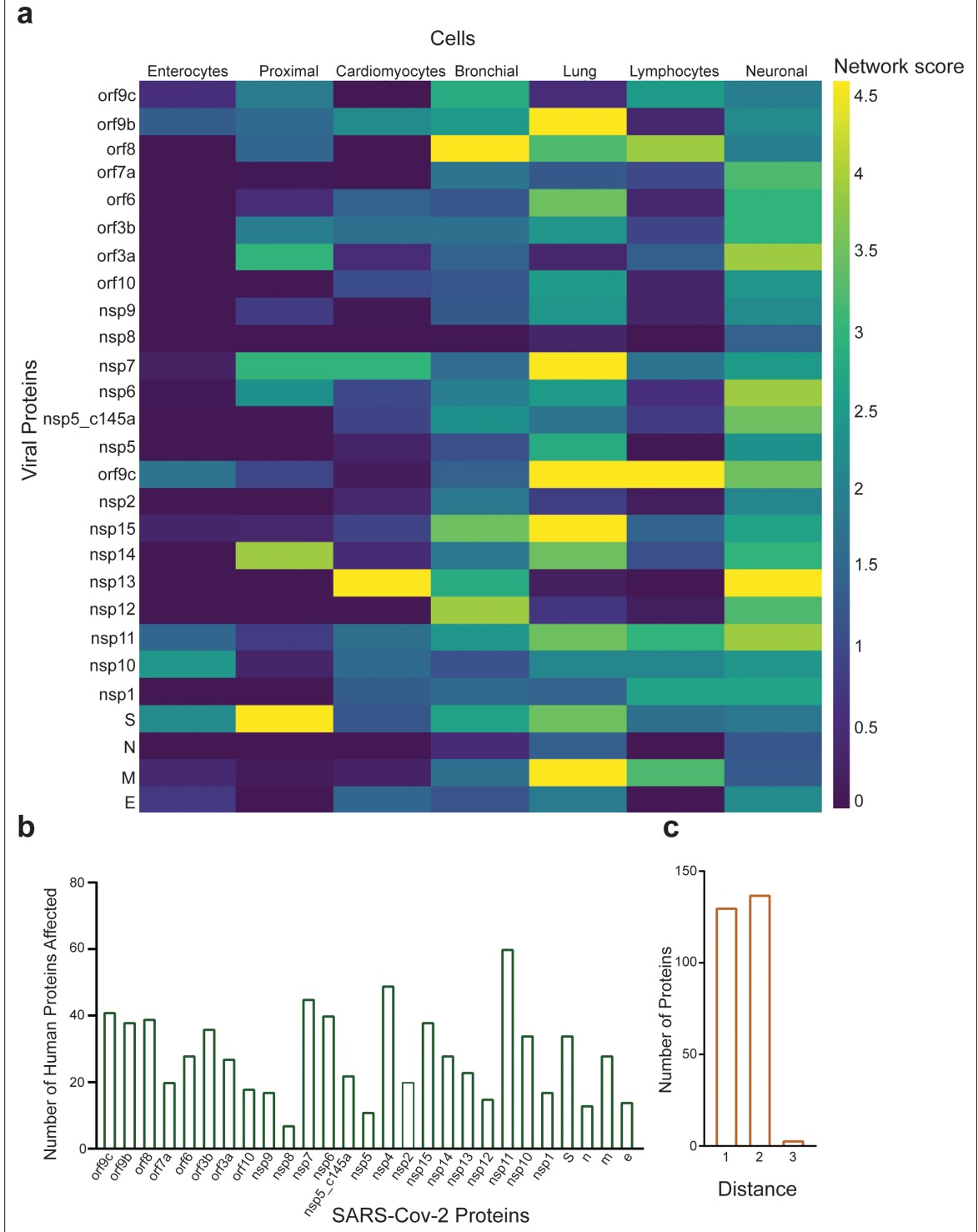

**Figure 7.** Protein identification using protein–protein interaction (PPI). (a) PPI results for the severe acute respiratory syndrome (SARS)-CoV-2 proteins that have a significant effect on the proteins presented in SI Table 1 for each system; (**b**) number of proteins affected by each SARS-CoV-2 protein, as calculated by PPI; (**c**) number of proteins with a specific distance factor from the viral proteins (also shown in SI Table 1).

The online version of this article includes the following figure supplement(s) for figure 7:

**Figure supplement 1.** Cardiomyocytes.

**Figure supplement 2.** Enterocytes.

**Figure supplement 3.** Lymphocytes.

*Figure 7 continued on next page*

expression, but also to confirm the data predicted by our model in which some TJ proteins can be more affected than others.

PPI analysis revealed a highly correlated effect of nsp7 and nsp13 on β-catenin in endothelial cells (*Figure 4b*; *Jung et al., 2020a*; *Lengfeld et al., 2017*). Interestingly, neither nsp2 nor nsp5_c145a affected a high number of proteins (*Figure 7b*), whereas nsp7 did, as identified by the network. Analyzing the repertoire of SARS-CoV-2 proteins, we see almost no effect of the structural proteins; rather, mostly nonstructural and open reading frame proteins affected HUVEC functionality, manifested as decreased barrier function and increased cytokine secretion (*Figures 2 and 3*). While the nonstructural proteins are mainly responsible for replicating viral RNA, the open reading frame proteins are related to counteraction with the host immune system; some of these are localized to the mitochondria and have been shown to alter the mitochondrial antiviral signaling pathway (*Miller et al., 2021*). We found that the proteins most affecting barrier function (decreased TEER and decreased CD31, β-catenin, cadherin-5, and ZO-1 expression) and cytokine response (IL-6 secretion and VWF expression) were nsp2, nsp5_c145a, and nsp7 (*Figure 2*; *Figure 3*; *Figure 6*); nsp7 forms a replication complex with nsp8 and nsp12 that is essential for viral replication and transcription (*Cowen et al., 2017*; *Peng et al., 2020a*). *Peng et al., 2020b* suggested that in the core polymerase complex nsp7–nsp8–nsp12, nsp12 is the catalytic subunit, and nsp7 and nsp8 function as cofactors. They further suggested that the mechanism of activation mainly involves the cofactors rather than the catalytic subunit (*Peng et al., 2020b*). This might explain why we saw mainly an effect of the cofactor proteins on endothelial cells and almost no effect of the catalytic subunit. Network interactions *Díaz, 2020* have shown that nsp7 has the most interactions with the host, suggesting a potential target for the treatment of COVID-19. Moreover, no mutations were found in nsp7 compared to nsp2 or nsp5_c145a (*Kaushal et al., 2020*), suggesting a conserved protein with a vital function in virus survival. The nsp13 protein has both helicase activity and 5' triphosphatase activity, which play an important role in mRNA capping. We saw a significant effect of nsp13 on barrier function, but hardly any effect on cytokine secretion. *Chen et al., 2020*, suggested functional complexation between nsp8 and nsp12, the RdRp (RNA-dependent RNA polymerase) replication complex, and nsp13. Given the fact that we observed a substantial effect of nsp7 – one of the proteins of the replication complex – and an effect of nsp13 on HUVEC barrier function, complexation of nsp13 with the replication complex might indicate an important role for this complex in the impaired functionality of the HUVECs, and therefore in the propagation of the disease, and the known vascular damage seen in COVID-19 patients. As suggested by our model, nsp13 seems to have a strong effect also on other types of vascular endothelial cells (*Figure 6*) as well as on all cell types (*Figure 7a*), positioning nsp13 as one of the main targets for disease treatment.

It is important to note that the comparison between the different endothelial cell types revealed exciting differences in the TJ protein expression, which correlate to the different properties of the different cell types (*Nakato et al., 2019*). One of the major differences was that some endothelial cell lines do not have cadherin at all (e.g., HAoEC), or very limited amount of cadherin (e.g., HPAEC, HUAEC, HGSVEC). Our model suggests that some endothelial types (e.g., HUVEC, HUAEC, HGSVEC, HCAEC) are more susceptible to the SARS-Cov-2 virus. It, therefore, suggests that the treatment of one type of endothelial cell might be different from another type but offers the PPI model as a tool for initial prediction. Overall, the combination of identifying the differences in the TJ protein expression between the different endothelial cells and the use of the PPI model enabled us to pinpoint the differences in susceptibility to the disease and to identify which specific proteins have the most significant effect.

Many studies have looked at the SARS-CoV-2 interaction with nonpulmonary/nonvascular tissues (e.g., neurons, hepatocytes, immune components such as lymphocytes, macrophages, etc.) (*Lee et al., 2021*), as pathological studies identified a viral effect on these tissues, despite their very limited amount, or lack of ACE2 receptors. To better understand how SARS-CoV-2 interacts with and affects

other tissues, we consolidated all of the proteins currently known to be affected by the virus into *Supplementary file 1A*. It is interesting to note that the most dominant SARS-CoV-2 proteins are nsp4, nsp11, and nsp7. *Davies et al., 2020*, identified the interaction of nsp2 with nsp4, both involved in endoplasmic reticulum (ER) calcium signaling and mitochondrial biogenesis. This suggests a new functional role in the host ER and mitochondrial organelle contact process and calcium homeostasis.

By now it is clear that vasculature plays a significant role in the physiological response to the virus. However, it is still unclear how the virus affects the vasculature, and if it can be found in the blood. This is a critical question, as it has significant consequences on the extent of the virus's ability to affect the vasculature. Current studies demonstrate that the pulmonary vasculature is significantly affected and is one of the dominant triggers for the pathologies mentioned above. However, involvement with the rest of the vasculature is still unclear, as is whether the virus can be found in an active form in the blood circulation (*Peng et al., 2020a*; *Chang et al., 2020*; *Orologas-Stavrou et al., 2020*; *Andersson et al., 2020*; *Escribano et al., 2020*; *Wang et al., 2020b*). Some studies suggest that even if there are traces of SARS-CoV-2 in the blood, it is not in an active form and cannot cause disease or a systemic response (*Andersson et al., 2020*). On the other hand, some studies suggest that SARS-CoV-2 can be found in the blood, and can induce the disease and cause both cellular and systemic dysfunction (*Peng et al., 2020a*; *Chang et al., 2020*; *Escribano et al., 2020*). While this question is beyond the scope of this work, it is important to note that if future studies do identify the active form of SARS-CoV-2 in human blood, then the implications of our findings will apply to this systemic response as well (*Ahmed et al., 2020*; *Park et al., 2020*).

As already noted, the pathology is probably a combination of multiple conditions and pathways activated by the different proteins. However, our findings might open new avenues for future therapeutics. Moreover, most of the proteins that were identified as affected by SARS-CoV-2 had a distance factor of at most three to the human and viral proteins. This coincides with the current dogma, whereby proteins that have a shorter distance between them are more likely to be affected.

While beneficial, our approach has two major limitations: (a) our inability to identify the effect of multiple proteins and (b) our neglect of the effect of the coronavirus structure and binding on the cellular response. The former point can be overcome by expressing combinations of different SARS-CoV-2 proteins. However, since the SARS-CoV-2 expresses 29 proteins, there are about $\sim 9 \times 10^{30}$ possible protein combinations. Therefore, we decided to focus on individual proteins and allow further studies to pursue any combinations of interest. Regarding the latter limitation, we did not include the coronavirus structure (including the ACE2 receptors) in this study, because many studies have already demonstrated the cellular response to this structure (*Chioh et al., 2020*; *Yang et al., 2020*; *Procko, 2020*), and how tissues that do not have significant ACE2 expression (neurons, immune components such as B and T lymphocytes, and macrophages) are affected by the virus remains an open question.

## Conclusions

Accumulating clinical evidence suggests that COVID-19 is a disease with vascular aspects. However, only a few studies have identified the specific role of each of the SARS-CoV-2 proteins in the cellular response leading to vascular dysfunctions. In this work, we characterized the endothelial response to each of 26 SARS-CoV-2 proteins and identified those that have the most significant effect on the barrier function. In addition, we used PPI network-based analysis to predict which of the endothelial proteins is most affected by the virus and to identify the specific role of each of the SARS-CoV-2 proteins in the observed changes in systemic protein expression. Overall, this work identified which of the SARS-CoV-2 proteins are most dominant in their effect on the physiological response to the virus. We believe that the data presented in this work will give us better insight into the mechanism by which the vasculature and the system respond to the virus, and will enable us to expedite drug development for the virus by targeting the identified dominant proteins.

## Materials and methods
### Generation of lentiviral SARS-CoV-2 plasmids

Plasmids encoding the SARS-CoV-2 open reading frames proteins and eGFP control were a kind gift of Nevan Krogan (Addgene plasmid #141367–141395). Plasmids were acquired as bacterial LB–agar stabs and used per the provider's instructions. Briefly, each stab was first seeded in LB agar (Bacto

Agar; BD Biosciences, San Jose, CA) in 10 cm plates. Then, single colonies were inoculated into flasks containing LB (BD Difco LB Broth, Lennox) and 100 µg/ml penicillin (Biological Industries, Beit HaEmek, Israel). Transfection-grade plasmid DNA was isolated from each flask using the ZymoPURE II Plasmid Maxiprep Kit (Zymo Research, Irvine, CA) according to the manufacturer's instructions.

## Lentivirus preparation

HEK293T cells (ATCC, Manassas, VA) were seeded in 10 cm cell culture plates at a density of $4 \times 10^6$ cells/plate. The cells were maintained in 293T medium composed of DMEM high glucose (4.5 g/l; Merck, Rahway, NJ) supplemented with 10 % fetal bovine serum (FBS; Biological Industries), $1 \times$ NEAA (Biological Industries), and 2 mM L-alanine–L-glutamine (Biological Industries, Israel). Lentiviral stocks, pseudo-typed with VSV-G, were produced in HEK293T cells as previously described (*Kutner et al., 2009*). Briefly, each of the pLVX plasmids containing the SARS-CoV-2 genes or EGFP for control were cotransfected with third-generation lentivirus helper plasmids at equimolar ratio; 48 hr later, the lentivirus-containing medium was collected and supernatant was clarified by centrifugation (500 g, 5 min) and filtration (0.45 µm, Millex-HV, Merck Millipore, Burlington, MA). All virus stocks were aliquoted and stored at –80 °C until thawed for subsequent use.

## Endothelial cell culture

HUVECs (C-12200, PromoCell GmbH, Heidelberg, Germany, tested negative for mycoplasma contamination) were used to test each viral protein's impact on vascular properties. After thawing, the HUVECs were expanded in low-serum endothelial cell growth medium (PromoCell) at 37 °C with 5 % $CO_2$ in a humidifying incubator, and used at passage p4–p6. Cells were grown to 80–90% confluence before being transferred to transparent polyethylene terephthalate Transwell supports (0.4 µm pore size, Greiner Bio-One, Austria) or a glass-bottom well plate (Cellvis, Mountain View, CA). Before seeding, the uncoated substrates were treated with Entactin-Collagen IV-Laminin (ECL) Cell Attachment Matrix (Merck) diluted in DMEM (10 µg/cm²) for 1 hr in the incubator. Then, the HUVECs were harvested using a DetachKit (PromoCell), were seeded inside the culture platforms at a density of 250,000 cells/cm², and grown for 3 days. Then viral infection with the different plasmids was performed and its impact on cell behavior was tested 3 days later.

## TEER measurement

The barrier properties of the endothelial monolayer were evaluated by TEER measurements, 3 and 4 days after viral infection. TEER was measured with the Millicell ERS-2 Voltohmmeter (Merck Millipore). TEER values ($\Omega$ cm²) were calculated and compared to those obtained in a Transwell insert without cells, considered as a blank, in three different individual experiments, with two inserts used for each viral protein.

## Immunofluorescence

HUVEC plated on glass-bottom plates were rinsed in phosphate buffered saline (PBS) and fixed in 4 % paraformaldehyde (Sigma-Aldrich, Rehovot, Israel) for 20 min at RT, 5 days after viral infection. ICC was carried out after permeabilization with 0.1 % Triton X-100 (Sigma-Aldrich, Rehovot, Israel) in PBS for 10 min at RT and blocking for 30 min with 5 % FBS in PBS. The following primary antibodies were applied overnight in PBS at 4 °C: rabbit anti-VWF (Abcam, Cambridge, UK), rabbit anti-CD31 (Abcam) against platelet endothelial cell adhesion molecule 1 (PECAM1), rabbit anti-β-catenin (Cell Signaling Technology, Danvers, MA), rabbit anti-cadherin-5 (Cell Signaling Technology, Danvers, MA), rabbit anti ZO-1 (Cell Signaling Technology, Danvers, MA), rabbit anti-occludin (Cell Signaling Technology, Danvers, MA). Cells were then washed three times in PBS and stained with the secondary antibody, anti-rabbit Alexa Fluor 488 (Invitrogen, Carlsbad, CA), for 1 hr at RT. After four washes with PBS, cells were incubated with Hoechst in PBS for 10 min at RT to stain the nuclei. After two washes with PBS, imaging was carried out using an inverted confocal microscope (Olympus FV3000-IX83) with suitable filter cubes and equipped with 20× (0.8 NA), 40× (0.95 NA), and 60× (1.42 NA) objectives. Image reconstruction and analysis were done using open-source ImageJ software (*Schindelin et al., 2012*).

## Network analysis

We scored the effect of each viral protein on selected human proteins using network propagation (*Cowen et al., 2017*). Specifically, a viral protein was represented by the set of its human interactors (*Hu et al., 2021*); each of these received a prior score, equal to 1 /n, where n is the size of the interactor set; these scores were propagated in a network of PPI (*Almozlino et al., 2017*). To assess the statistical significance of the obtained scores, we compared them to those computed on 1000 randomized networks that preserve node degrees. The PPI score was then compared versus the other random networks (this is empirical p-value). p-Values were adjusted for multiple testing using Benjamini–Hochberg FDR approach. For display purposes, the plotted p-value is the negative log of the p-value, which means numbers are non-negative and the higher is the more significant.

## Quantitative ELISA for IL-6

ELISA was performed on conditioned medium of infected HUVEC 3 days postinfection, according to the manufacturer's recommendations (PeproTech Rehovot, Israel).

## Statistical analysis

The results are presented as mean ± SD, unless otherwise indicated. Statistically significant differences among multiple groups were evaluated by F-statistic with two-way ANOVA, followed by the Holm–Sidak test for multiple comparisons (GraphPad Prism 8.4.3). The difference between the two data sets was assessed and $p < 0.05$ was considered statistically significant.

## Acknowledgements

BMM was supported by the Azrieli Foundation, Israel Science Foundation (ISF grant: 2248/19), ERC SweetBrain 851765, TEVA, The Aufzien Family Center for the Prevention and Treatment of Parkinson's Disease at Tel Aviv University, Zimin, Israel Ministry of Science and Technology (Grant No. 3–17351), and TCCP. UA was supported by the Israel Science Foundation (ISF grant 953/16), TEVA, The Aufzien Family Center for the Prevention and Treatment of Parkinson's Disease at Tel Aviv University, the German Research Foundation (DFG) (NA: 207/10–1) and the Taube/Koret Global Collaboration in Neurodegenerative Diseases. RS was supported by the Israel Science Foundation (ISF grant 2417/20), within the Israel Precision Medicine Partership program. The work of YN, AE, and KI was supported by European Research Council Consolidator Grant OCLD (project no. 681870).

## Additional information

### Funding

| Funder | Grant reference number | Author |
|---|---|---|
| Israel Science Foundation | 2248/19 | Rossana Rauti<br>Yael Leichtmann-Bardoogo<br>Rina Tamir<br>Victoria Miller<br>Tal Babich<br>Kfir Shaked<br>Ben Meir Maoz |
| Azrieli Foundation | | Rossana Rauti<br>Yael Leichtmann-Bardoogo<br>Ben Meir Maoz |
| Horizon 2020 | SweetBrain 851765 | Rossana Rauti<br>Yael Leichtmann-Bardoogo<br>Ben Meir Maoz |

| Funder | Grant reference number | Author |
|---|---|---|
| Aufzien Family Center for the Prevention and Treatment of Parkinson's Disease | | Rossana Rauti<br>Meishar Shahoha<br>Yael Leichtmann-Bardoogo<br>Eyal Paz<br>Rina Tamir<br>Victoria Miller<br>Tal Babich<br>Kfir Shaked<br>Uri Ashery<br>Ben Meir Maoz |
| Deutsche Forschungsgemeinschaft | 207/10-1 | Rami Nasser<br>Avner Ehrlich<br>Konstantinos Ioannidis<br>Yaakov Nahmias<br>Roded Sharan |
| Teva Pharmaceutical Industries | | Yael Leichtmann-Bardoogo<br>Ben Meir Maoz |
| Zimin | | Yael Leichtmann-Bardoogo<br>Ben Meir Maoz |
| Ministry of Science and Technology, Israel | 3-17351 | Rossana Rauti<br>Yael Leichtmann-Bardoogo<br>Ben Meir Maoz |
| TCCP | | Uri Ashery<br>Ben Meir Maoz |
| Israel Science Foundation | 953/16 | Rina Tamir<br>Victoria Miller<br>Tal Babich<br>Kfir Shaked<br>Ben Meir Maoz |

The funders had no role in study design, data collection and interpretation, or the decision to submit the work for publication.

### Author contributions

Rossana Rauti, Conceptualization, Formal analysis, Methodology, Project administration, Writing – original draft; Meishar Shahoha, Conceptualization, Data curation, Investigation, Methodology, Visualization; Yael Leichtmann-Bardoogo, Conceptualization, Data curation, Investigation, Methodology, Visualization, Writing – original draft; Rami Nasser, Roded Sharan, Formal analysis, Software; Eyal Paz, Methodology, Writing - review and editing; Rina Tamir, Victoria Miller, Tal Babich, Kfir Shaked, Formal analysis, Investigation; Avner Ehrlich, Konstantinos Ioannidis, Yaakov Nahmias, Methodology, Resources; Uri Ashery, Funding acquisition, Investigation, Supervision, Writing – original draft; Ben Meir Maoz, Conceptualization, Funding acquisition, Project administration, Supervision, Visualization, Writing – original draft

### Author ORCIDs

Rossana Rauti  http://orcid.org/0000-0001-8569-0810
Meishar Shahoha  http://orcid.org/0000-0001-5947-484X
Uri Ashery  http://orcid.org/0000-0001-6338-7888
Ben Meir Maoz  http://orcid.org/0000-0002-3823-7682

### Decision letter and Author response

Decision letter https://doi.org/10.7554/eLife.69314.sa1
Author response https://doi.org/10.7554/eLife.69314.sa2

---

## Additional files

### Supplementary files

• Supplementary file 1. Protein change by SARS-Cov-2. (A) Documented change by severe acute respiratory syndrome (SARS)-Cov-2. (B) Significant target for each viral protein.

• Transparent reporting form

• Source data 1. The source data file contains all the data that was used for *Figure 2*, *Figure 3*, *Figure 5*, and *Figure 2—figure supplement 1b*.

## Data availability

All data generated or analysed during this study are included in the manuscript and supporting files. The custom scripts available in GitHub: https://github.com/raminass/covid_networks, (copy archived at https://archive.softwareheritage.org/swh:1:rev:b239ae7e0e72b722beb6d694436068541ea28dbb).

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
