## [Decision Letter]

**Acceptance summary:**

In their manuscript, Rauti, Shahoha et al., sought to identify the effect of SARS-CoV2 proteins on endothelial functions. They systematically transduced HUVECs cultured on a transwell with various SARS-CoV2 proteins from three classes including non-structural and accessory proteins. They have assessed the effect of separate overexpression of each of 26 proteins, and tested their effect on 26 of the 29 proteins in the SARS-CoV2 genome. The authors found that overexpression of some proteins had stronger effect on the barrier functions on HUVECs compared to other proteins. Using a PPI network analysis, they predicted endothelial proteins that may be affected by the viral protein, which may potentially mediate this effect. The findings add to the understanding how SARS-CoV2 may negatively affect endothelial cell homeostasis, which may contribute to vascular and thrombotic complications associated with a severe course of the disease in patients diagnosed with COVID-19.

**Decision letter after peer review:**

Thank you for submitting your article "Effect of SARS-CoV-2 proteins on vascular permeability" for consideration by *eLife*. Your article has been reviewed by 2 peer reviewers, and the evaluation has been overseen by a Reviewing Editor and Matthias Barton as the Senior Editor. The following individuals involved in review of your submission have agreed to reveal their identity: Francesco Pasqualini (Reviewer #1); Gad Vatine (Reviewer #2).

Essential revisions:

1) The authors provide putative virus-host protein-protein interaction networks that could be very valuable for future translational efforts. However, it would be best to validate at least one of those on the HUVECS/TEER assay. Would it be possible to rescue endothelial dysfunction acting (pharmacologically or genetically) on any of the identified host proteins downstream of the viral protein?

2) The authors provide virus-host protein interaction networks for endothelial cells of various organs. This, again, could be a very valuable target identification exercise for future pharmacological work but it would need to be validated. Can the rescue experiment discussed above be replicated using the TEER assay, organ-specific endothelial cells, and pharmacological or genetic means to normalize the expression of host proteins downstream of the viral protein in the organ-specific PPI network?

3) More validation data will help but it might be worth revising the discussion as well, especially as the new validation data may make parts of the current Discussion redundant.

4) Page 4, line 98 – "Permeability was measured via trans-epithelial-endothelial 99 electrical resistance (TEER), a standard method that identifies changes in impedance values". TEER is a method for assessing barrier functions, and it is related to permeability, but permeability should be measured by fluorescent paracellular assays. These should be separated.

5) While significant, the changes in TEER are very small. The authors should specify in the figure legend, which statistical tests have been applied, and should specify that they have accounted for multiple comparison tests. Also, changes in TEER over time should be showed.

6) In order to avoid conclusions that may be linked to direct regulation on CD31, changes in protein expression should be made on additional proteins that are related to tight junctions such as ZO1.

7) Overall, all experiments were performed on a single source of in vitro human endothelial cells. The authors should discuss possibilities for additional endothelial cell models that could potentially be used for future studies, and discuss the advantages and disadvantages of HUVECs.

8) The authors develop a mathematic model to identify interactions of the SARS-CoV2 proteins with tissue targets. While this represents a novel an important approach, authors should show a validation of the model to prove its validity. If this cannot be applied in the HUVECs system, the authors should explain its limitations and at least discuss what possible models could be used to validate it in the future.

9) Since the authors identify WNT signaling as a key pathway, the authors cold consider manipulating the WNT pathway in HUVECs cells as a possible treatment for rescuing this effect.*Reviewer #1:*

The hypothesis of the authors is that non-spike SARS-CoV-2 proteins are responsible for the increased vascular permeability observed in COVID-19 patients during the primary infection and, possibly, long after it is resolved. To identify which protein(s) might be responsible, they used lentiviruses to express 26/29 SARS-CoV-2 proteins in human endothelial cells (HUVECS) using trans-epithelial-endothelial electrical resistance (TEER), a standard assessment of vascular permeability. Among the viral proteins that directly increased vascular permeability, the authors identified a subset (nsp2,29 nsp5_c145a (catalytic dead mutant of nsp5) and nsp7) capable of inducing broader endothelial dysfunction (downregulated CD31 and upregulated VWF and IL-6). Finally, they used protein-protein interaction analysis to semi-quantitatively speculate how these viral proteins may lead to endothelial dysfunction and increased vascular permeability in various human organs.

Strengths

The use of a library of LV to test each individual SARS-CoV-2 protein is interesting. While hard to use combinatorially, being able to isolate the influence of each protein will be important in understanding the secondary effect of COVID-19 infections (e.g., long COVID).

The combination of imaging and TEER to screen how each viral protein changes endothelial structure, function, or both.

The use of bio-informatics to relate the raw findings from the cellular assay with broader translational implications, namely through which pathways may viral protein end up affecting vascular permeability and endothelial dysfunction.

Weaknesses

I feel like the paper is missing two key additional validations.

First, the authors provide putative virus-host protein-protein interaction networks that could be very valuable for future translational efforts. However, it would be best to validate at least one of those on the HUVECS/TEER assay. Would it be possible to rescue endothelial dysfunction acting (pharmacologically or genetically) on any of the identified host proteins downstream of the viral protein?

Second, the authors provide virus-host protein interaction networks for endothelial cells of various organs. This, again, could be a very valuable target identification exercise for future pharmacological work but it would need to be validated. Can the rescue experiment discussed above be replicated using the TEER assay, organ-specific endothelial cells, and pharmacological or genetic means to normalize the expression of host proteins downstream of the viral protein in the organ-specific PPI network?*Reviewer #2:*

In their manuscript, Rauti, Shahoha et al., sought to identify the effect of SARS-CoV2 proteins on endothelial functions. They systematically transduced HUVECs cultured on a transwell with various SARS-CoV2 proteins from three classes including non-structural and accessory proteins. They have assessed the effect of separate overexpression of each of 26 proteins, and tested their effect on 26 of the 29 proteins in the SARS-CoV2 genome. The authors found that overexpression of some proteins had stronger effect on the barrier functions on HUVECs compared to other proteins. Using a PPI network analysis, they predicted endothelial proteins that may be affected by the viral protein, which may potentially mediate this effect.

The novelty of the study is in directly testing the response of endothelial cells to the virus. Previous reports focused on the phenotype in patients, or using animal models without specifying the exact proteins involved in the dysfunction. Gaining knowledge on the molecular mechanisms, especially the protein-protein interaction pathways that lead to tissue dysfunction, may help understanding the pandemic a potentially help develop treatments.

---

## [Author Response]

Essential revisions:1) The authors provide putative virus-host protein-protein interaction networks that could be very valuable for future translational efforts. However, it would be best to validate at least one of those on the HUVECS/TEER assay. Would it be possible to rescue endothelial dysfunction acting (pharmacologically or genetically) on any of the identified host proteins downstream of the viral protein?

We would like to thank the referee for his suggestion. We agree that it is important to show the validity of the model before using it on other proteins. Following the referee’s suggestion, we have performed a series of experiments to validate the model (new Figure 5). Our new data present high correlation between the PPI prediction and the experimental data.

2) The authors provide virus-host protein interaction networks for endothelial cells of various organs. This, again, could be a very valuable target identification exercise for future pharmacological work but it would need to be validated. Can the rescue experiment discussed above be replicated using the TEER assay, organ-specific endothelial cells, and pharmacological or genetic means to normalize the expression of host proteins downstream of the viral protein in the organ-specific PPI network?

Following the referee’s suggestion to examine different organ-specific endothelial cells, we have made a unique comparison of the expression level of the different endothelial proteins in 9 different vascular endothelial cell types (see new Table 2).

Although we assumed that most proteins will be detected in these lines, we were surprised to learn that in some cell lines, like the HAoEC, all cadherin proteins (Cadherin 2-5) do not express while in others Catenin d is not expressed while in other most ZO proteins are missing. We were surprised to learn that such a comparison was never published, making this publication also unique in that way.

After we validated the PPI network, we used it to examine how the specific SARS-Cov-2 proteins affect the organ-specific endothelial cells (for 9 organ-specific vascular endothelial cells). Our results presented in new Figure 6 predict that not only the HUVEC permeability properties are impaired by the SARSCov-2 proteins, but other vascular endothelial cells are strongly affected as well. Interestingly, we see that different endothelial cells response differently to SARS-Cov-2, mainly due to the different expression of the tight junctions proteins.

3) More validation data will help but it might be worth revising the discussion as well, especially as the new validation data may make parts of the current Discussion redundant.

The discussion was revised according to the new validation.

4) Page 4, line 98 – "Permeability was measured via trans-epithelial-endothelial 99 electrical resistance (TEER), a standard method that identifies changes in impedance values". TEER is a method for assessing barrier functions, and it is related to permeability, but permeability should be measured by fluorescent paracellular assays. These should be separated.

We thank the author for this comment. The text was revised to better clarify this point.

5) While significant, the changes in TEER are very small. The authors should specify in the figure legend, which statistical tests have been applied, and should specify that they have accounted for multiple comparison tests. Also, changes in TEER over time should be showed.

In this work we used ANOVA-multiple comparison test to account for the multiple comparison among different groups. This was described in the method section and per the reviewer request, it was also added to the figure legend.

The TEER dynamic is highly dependent on the cell density, age, and condition. Per the reviewer request, we added the TEER values of the cells in multiple time points (before the infection, 3 and 4 days after the infection (new Figure S1)).

6) In order to avoid conclusions that may be linked to direct regulation on CD31, changes in protein expression should be made on additional proteins that are related to tight junctions such as ZO1.

We would like to thank the referee for this comment. We have performed additional set of experiments examining also the effect on ZO1 (and other 3 proteins, B-Catenin, Cadherine 5 and Occludin). The new data, now in new Figure 5, show that the viral proteins that showed a significant decrease in the TEER values as well as in the CD31 intensity, significant affect the expression of these other tight-junctions proteins.

7) Overall, all experiments were performed on a single source of in vitro human endothelial cells. The authors should discuss possibilities for additional endothelial cell models that could potentially be used for future studies, and discuss the advantages and disadvantages of HUVECs.

As we discussed in “point 2”, we created a table which compares 9 organ-specific endothelial cells. We added the results as Table 2 and new Figure 6**.**

We also revised the text accordingly.

8) The authors develop a mathematic model to identify interactions of the SARS-CoV2 proteins with tissue targets. While this represents a novel an important approach, authors should show a validation of the model to prove its validity. If this cannot be applied in the HUVECs system, the authors should explain its limitations and at least discuss what possible models could be used to validate it in the future.

As we discussed in “point 2”, we created a table which compares 9 organ-specific endothelial cells. We added the results as Table 2 and new Figure 6**.**

We also revised the text accordingly.

9) Since the authors identify WNT signaling as a key pathway, the authors cold consider manipulating the WNT pathway in HUVECs cells as a possible treatment for rescuing this effect.

WNT was suggested as a relevant pathway due to its known contribution to BBB impairment in multiple sclerosis and other pathologies (Jung, Y., et al. ACS Infect. Dis. (2020); Lengfeld, J. E.,et al. Proc Natl Acad Sci (2017)). In addition, the ßcatenin is part of the WNT signaling, and we showed the validation of its effect by the SARS-Cov-2, i.e. decrease in expression caused by the virus infection. But since it is not the main claim of the paper, (the contribution of the WNT to the BBB impairment due to the SARS-Cov-2) we removed this point from the discussion. Manipulation of the WNT signaling pathway is beyond the scope of this paper.